# Effects of Tea Polyphenol and Its Combination with Other Antioxidants Added during the Extraction Process on Oxidative Stability of Antarctic Krill *(Euphausia superba)* Oil

**DOI:** 10.3390/foods11233768

**Published:** 2022-11-23

**Authors:** Ziwei Wang, Fujun Liu, Ying Luo, Xiangbo Zeng, Xuechen Pei, Guanhua Zhao, Min Zhang, Dayong Zhou, Fawen Yin

**Affiliations:** 1School of Food Science and Technology, Dalian Polytechnic University, Dalian 116034, China; 2Liao Fishing Group Limited Company, Dalian 116000, China; 3National Engineering Research Center of Seafood, Dalian Polytechnic University, Dalian 116034, China; 4Collaborative Innovation Center of Seafood Deep Processing, Dalian 116034, China

**Keywords:** antarctic krill, oil, antioxidant, tea polyphenol, composite antioxidants, ethanol

## Abstract

Antarctic krill (*Euphausia superba*) oil contains high levels of marine omega-3 long-chain polyunsaturated fatty acids (n-3 LC-PUFA), including eicosapentaenoic acid (EPA) and docosahexaenoic acid (DHA). In industrial production, krill oil is usually extracted from krill meals by using ethanol as a solvent. Water in the krill meal can be easily extracted by using ethanol as an extraction solvent. During the extraction process, the EPA and DHA are more easily oxidized and degraded when water exists in the ethanol extract of krill oil. Based on the analysis of peroxide value (POV), thiobarbituric acid-reactive substances (TBARS), fatty acid composition, and lipid class composition, the present study indicated that the composite antioxidants (TP-TPP) consist of tea polyphenol (TP) and tea polyphenol palmitate (TPP) had an excellent antioxidant effect. By contrast, adding TP-TPP into ethanol solvent during the extraction process is more effective than adding TP-TPP into krill oil after the extraction process.

## 1. Introduction

Antarctic krill (*Euphausia superba*, *E. superba*) is a plentiful source of high-quality protein, with a protein level that is believed to be between 60–65% of its dry weight [1]. The protein in krill is a complete protein, which includes all nine essential amino acids that humans need [1]. In addition to protein, oil is another essential nutritional component of krill. Krill oil is a growing source of marine omega-3 long-chain polyunsaturated fatty acids (n-3 LC-PUFAs), including eicosapentaenoic acid (EPA) and docosahexaenoic acid (DHA). Ulven et al. reported that most of the n-3 LC-PUFA in fish oil are integrated into triglycerides [2]. By contrast, n-3 LC-PUFA in krill oil is mainly incorporated into phospholipids (PL). Owing to structural differences, it was expected that krill oil would have a higher bioavailability of n-3 LC-PUFA than fish oil [3]. Moreover, krill oil contains a lot of naturally occurring antioxidants, such as astaxanthin, which may be the cause of its biological properties [4]. So far, numerous reports have confirmed the health benefits of krill oil, including lowering hepatic steatosis and preventing hyperglycemia [5], enhancing memory and cognitive performance [6], reducing inflammation and oxidative stress [7], and heart protection [8].

It is widely known that during the heat processing or storage of oils, lipid oxidation can be accelerated and can produce specific odors and flavors [9]. Numerous aldehydes and ketones affect the quality and safety of oils in addition to their sensory characteristics [10]. It was discovered that aldehydes and ketones were produced by the breakdown of PUFAs in oils [11]. Therefore, krill oil is very vulnerable to oxidation due to its high PUFA content (particularly EPA and DHA). For example, Yin et al. reported that after 8 weeks of storage at 40 °C in light without oxygen, a decrease was observed in the relative percentage of PUFA [12]. There was a 2.2% drop in the value, which went from 29.82 to 27.62%. Moreover, Thomsen et al. reported that following 21 days of storage at 40 °C, the content of a few secondary volatile lipid oxidation products increased significantly (octanal, 107 to 345 ng/g; benzaldehyde, 153,792 to 324,042 ng/g) [13].

Antioxidants are compounds capable of slowing down the oxidation rate of lipids [14], which can effectively prolong the shelf life of edible oils. According to Choe and Min, the antioxidant mechanism of antioxidant has been reported to include scavenging free radicals, chelating metal ions, and quenching singlet oxygen [14]. To improve the stability, oil products are usually added with mixed antioxidants with different antioxidant mechanisms. These antioxidants usually have complementary and synergistic effects [15,16]. For example, Omar et al. reported that higher antioxidant activity in flaxseed oil was found when 100 mg/kg TBHQ and 200 mg/kg polyphenols were combined [17]. Moreover, Rudnik and Winiarska revealed that the antioxidant stability of microalgal DHA-rich oil could be improved by a combination of rosemary extract (RE), vitamin E (VE), and ascorbyl palmitate (AP) [18].

Currently, krill oil is extracted by organic solvents in industrial production [19]. Obviously, oils are easily oxidized and degraded in the procedures of settling (contact with air) and evaporation (relatively high temperature). By contrast, the extraction solvent of vegetable oil is No. 6 solvent (n-hexane), while the extraction solvent of krill oil is ethanol [20]. Water in the krill meal can be easily extracted by using ethanol as an extraction solvent. During the extraction process, the EPA and DHA are more easily oxidized and degraded when water exists in the ethanol extract of krill oil. Therefore, adding antioxidants to the extraction solvent during the oil extraction process may possibly inhibit oil oxidation.

Given this, this study aimed to select the most effective single antioxidant and composite antioxidant among vitamin C (VC), tea polyphenol (TP), ascorbyl palmitate (AP), vitamin E (VE), antioxidant of bamboo leaves (AOB), tea polyphenol palmitate (TPP), rosemary extract (RE) and their binary mixtures, as well as compare the accelerated oxidative stability of krill oils added with antioxidants at different time points (during or after the extraction process). Especially the composite antioxidant consists of the selected single antioxidant (the best antioxidation effect) with other commonly used antioxidants (AP, VC, VE, AOB, RE, and TPP). This study will provide a better understanding of protecting Antarctic krill oil from oxidation and afford the basis for extending the shelf-life of krill oil products.

## 2. Materials and Methods

### 2.1. Materials and Chemicals

Krill meal was purchased from Liao Fishing Group Limited Company (Dalian, China). Food-grade vitamin C (VC) and tea polyphenol (TP) were purchased from Jianming Technologies Co., Ltd. (Zhuhai, China). Food-grade vitamin E (VE) and ascorbyl palmitate (AP) were purchased from Aladdin Reagent Co., Ltd. (Beijing, China). Food grade antioxidant of bamboo leaves (AOB) was purchased from Aikon Biopharmaceutical R&D Co., Ltd. (Nanjing, China). Food-grade tea polyphenol palmitate (TPP) was purchased from Guangzhou shengtong trading Co., Ltd. (Guangzhou, China). Food-grade rosemary extract (RE) was purchased from Henan Yuzhong biology science and technology Co., Ltd. (Zhengzhou, China). Ethanol was purchased from Tianjin Damao Chemical Reagent Co., Ltd. (Tianjin, China).

### 2.2. The Preparation of Krill Oil Samples Added with Antioxidants during the Extraction Process

A total of 18 g of krill meal was weighed, and the lipid was extracted using 90 mL of solvent (ethanol) added with the single antioxidant or the composite antioxidant at 25 °C for 30 min. Sitting in the dark for 10 min, the mixture was filtered using a Buchner funnel. Subsequently, the filtrate was collected, and the ethanol in filtrate was removed through rotary evaporation at 30 °C. Thus, the krill oil samples added with the single antioxidants or the composite antioxidants during the extraction process were obtained.

Especially based on the oil extraction rate, the single antioxidant (VC, TP, AP, VE, AOB, TPP, or RE) was added to the ethanol at its maximum allowable quantity (maq) allowed by Chinese Standard GB 2760-2014 [21]. The maq values of VC, TP, AP, VE, AOB, TPP, and RE were 0.2, 0.6, 0.2, 0.4, 0.5, 0.6, and 0.7 g/kg oil, respectively. As for the composite antioxidant (TP-VC, TP-AP, TP-VE, TP-AOB, TP-TPP or TP-RE) comprised of TP and the other antioxidant (VC, AP, VE, AOB, TPP or RE), the TP, VC, AP, VE, AOB, TPP or RE was added to the ethanol at its one half of maq allowed by Chinese Standard GB 2760-2014 [21].

### 2.3. The Preparation of Krill Oil Samples Added with Antioxidants after the Extraction Process

According to the above extraction steps, krill oil was extracted from krill meal by using ethanol without adding any antioxidant as extraction solvent. In order to investigate the antioxidant effect of the antioxidants added at different time points, the single antioxidants or the composite antioxidants were added directly to krill oils, respectively.

Especially according to Chinese Standard GB 2760-2014 [21], the TP and the TP-TPP were added to the oil at their maximum allowable quantity: TP (400 mg/kg), TP-TPP (TP, 200 mg/kg; TPP, 300 mg/kg). Thus, the krill oil samples added with the single antioxidants or the composite antioxidants after the extraction process were obtained.

### 2.4. Accelerated Storage Experiment

The krill oil samples added with antioxidants during or after the extraction process were taken at regular intervals of 2 days until 8 days during an accelerated storage experiment at 60 °C.

### 2.5. Peroxide Value

The peroxide value (POV) of krill oil samples was measured according to a previous method [22]. In short, krill oils (0.01 g) were dissolved in 1.5 mL of dichloromethane: 95% ethanol (3:2, *v*/*v*). Then 5 mM aqueous ferrous ammonium sulfate (100 μL), 1 M methanolic XO (200 μL), and 0.25 M methanolic H_2_SO_4_ (200 μL) were added. One mL of distilled water was added to the reaction after it had been left at room temperature and in the dark for 30 min. Then centrifuged at 4000× *g* for 5 min. Took 200 uL of the mixture’s upper layer and measured the absorbance at 560 nm. The POV was determined using a CHP calibration curve.

### 2.6. Thiobarbituric Acid Reactive Substances

Using the method in [23], the Thiobarbituric acid reactive substances (TBARS) of krill oil samples were performed. Briefly, krill oil (0.1 g) was mixed equally with mixed liquor (2.5 mL), which included distilled water (196 mL), concentrated hydrochloric acid solution (4.17 mL), thiobarbituric acid (0.75 g) and trichloroacetic acid (30 g). The above mixture was heated for ten minutes in a bath of boiling water. After cooling and centrifuging at 3000× *g* for 10 min, took 200 uL of the mixture’s upper layer and measured the absorbance at 532 nm. The malondialdehyde concentration was converted to TBARS number as follows: TBARS (ppm) = sample A_532_ × 2.77.

### 2.7. Fatty Acid Composition

According to our previous method [24], fatty acid methyl esters (FAMEs) were prepared by methylation. In short, lipid sample (5 mg) was mixed with an internal standard solution (200 µL) of 1 mg/mL tridecanoyl glyceride dissolved in chloroform. Then 0.5 M NaOH-CH_3_OH (2 mL) was added. Next, refluxed in a water bath at 80 °C for 5 min, and then BF_3_-methanol solution (2 mL; 14%, *w*/*w*) was added for 2 min through a condenser. Subsequently, the mixture was cooled and extracted with hexane (1.5 mL). Before undergoing gas chromatographic (GC) analysis, hexane containing FAMEs was put through a 0.22 μm filter. FAME separation was performed by using a Supelco SP 2560 capillary column (100 m × 0.25 mm, 0.2 μm). The injection volume was 1 μL with a split ratio of 20:1, and the injector temperature was set as 220 °C. The FID temperature was set as 260 °C, and the constant carrier gas (N_2_) flow was set as 2.0 mL/min. The heating procedure is as follows: 120 °C for 9 min; increasing (20 °C/min) to 200 °C and held for 5 min; increasing (3 °C/min) to 230 °C and held for 10 min. All fatty acids were identified by comparing their retention times with those of the standards [25].

### 2.8. Lipid Class Composition

According to the previous study [26], the lipid class composition of krill oil samples was determined by using the Iatro-scan MK-6S thin layer chromatography-flame ionization detection (TLC-FID) Analyzer (Iatron Inc., Tokyo, Japan). Krill oil (0.02 g) was dissolved in chloroform (2 mL). The above lipid sample (1 μL) was spotted onto a quartz rod (SIII Chromarods, Iatron Inc., Tokyo, Japan), and the elution was performed with formic acid/diethyl ether/n-heptane (*v*/*v*/*v*, 0.3:28:42) for 20 min. Before scanning each Chromarod with FID, Chromarods were dried at 60 °C. After data collection and processing, comparison of migration distance with reliable standards was used to identify the lipid. By dividing the peak area of the separated lipid by the sum of the peak areas of all the separated lipids, the lipid class compositions of triglyceride (TG), free fatty acid (FFA), diglyceride (DG), cholesterol (Cho), monoglyceride (MG) and phospholipid (PL) were obtained.

### 2.9. Statistical Analysis

The experiments mentioned above were carried out three times, and the results were provided as mean ± standard deviation (SD). The data were analyzed by SPSS (version 26, IBM Corp., Armonk, NY, USA), then one-way analysis of differences was used to assess the difference between means (*p* < 0.05).

## 3. Results

### 3.1. Selection of the Most Effective Single Antioxidant Added during the Extraction Process

POV was chosen to determine the amounts of hydroperoxides formed during the extraction process of krill oils (Figure 1A). The POV values of krill oils added with single antioxidants (vitamin C (VC), vitamin E (VE), tea polyphenol (TP), ascorbyl palmitate (AP), tea polyphenol palmitate (TPP), rosemary extract (RE), and antioxidant of bamboo leaves (AOB)) were significantly lower than that of the control group (Con) without adding any antioxidants, showing that these antioxidants could significantly retard the primary oxidation of krill oils (*p* < 0.05). Apparently, based on the POV, the antioxidant efficiency of TP and TPP was greater than that of the other antioxidants. The order of inhibitory ability was: TP, TPP > RE > AOB > VE > VC > AP > Con (*p* < 0.05).

TBARS was used to measure the formation of secondary oxidation products during the extraction process of krill oils (Figure 1B). The TBARS values of krill oils added with single antioxidants were significantly lower than that of the control group, showing that these antioxidants could significantly retard the secondary oxidation of krill oils (*p* < 0.05). Obviously, based on the TBARS, the antioxidant efficiency of TP and TPP was greater than that of the other antioxidants. The order of inhibitory ability was: TPP, TP > RE > AOB > VE > VC > AP > Con (*p* < 0.05).

The fresh krill oils contained 23.04% of PUFA, 27.22% of MUFA, and 49.73% of SFA (Table 1). Furthermore, the main PUFA, DHA, and EPA make up 4.60% and 10.60% of all fatty acids, respectively. In order to further confirm the above-mentioned results, the changing trends in the fatty acid composition of krill oils added with single antioxidants were measured. The PUFA values of krill oils added with single antioxidants were significantly higher than that of the control group, while SFA and MUFA were lower than those of the control group (*p* < 0.05). Obviously, all the added single antioxidants could significantly inhibit the decrease in PUFA levels during the extraction process (*p* < 0.05). Similarly, TP and TPP exerted a higher ability to inhibit the oxidation loss of PUFA than others during the extraction process. The order of inhibitory ability was: TPP > TP > RE > AOB > VE > VC > AP > Con (*p* < 0.05).

The above results clearly indicated that, during the extraction process of krill oils, TP and TPP exerted the best antioxidant effectiveness among the seven single antioxidants. Compared with TPP, as a kind of natural antioxidant, TP has been extensively used in the food and feed industry. Thus, TP was selected to combine with VC, AOB, VE, AP, TPP, and RE, respectively, to form composite antioxidants to further enhance the oxidative stability of krill oils during the extraction process.

### 3.2. Selection of the Most Effective Composite Antioxidant Added during the Extraction Process

The POV values of krill oils added with TP-VC, TP-AP, TP-VE, TP-AOB, TP-TPP, TP-RE (the binary mixtures comprised of TP and one of the other six antioxidants (VC, AP, VE, AOB, TPP, and RE)) and TP, were significantly lower than that of the control group without adding any antioxidants (*p* < 0.05) (Figure 1C). The result indicated that TP-TPP and single TP had the most excellent antioxidant effect. The order of inhibitory ability was: TP-TPP, TP > TP-RE > TP-AOB > TP-AP > TP-VE > TP-VC > Con (*p* < 0.05).

The TBARS values of krill oils added with TP-VC, TP-AP, TP-VE, TP-AOB, TP-TPP, TP-RE, and TP were significantly lower than that of the control group (*p* < 0.05) (Figure 1D). The result indicated that TP-TPP had the most excellent antioxidant effect, then the single TP. The order of inhibitory ability was: TP-TPP > TP > TP-RE > TP-AOB > TP-AP > TP-VE > TP-VC > Con (*p* < 0.05).

In order to further confirm the above-mentioned results, the changing trends in the fatty acid composition of krill oils added with composite antioxidants were measured (Table 2). The SFA and MUFA values of krill oils added with antioxidants were lower than that of the control group, while PUFA were significantly higher than those of the control group (*p* < 0.05). Similar to the results of POV and TBARS, all the added antioxidants, especially TP-TPP and TP, could significantly inhibit the decline of PUFA during the extraction process. The order of inhibitory ability was: TP-TPP > TP > TP-RE > TP-AOB > TP-AP > TP-VE > TP-VC > Con (*p* < 0.05).

The above results clearly indicated that, during the extraction process of krill oils, TP and TP-TPP were the most effective single and composite antioxidants, respectively. However, adding the antioxidants at different time points of processing and storage may influence the oxidative stability of oil products. Given this, the effects of TP and TP-TPP added during or after the extraction process on the oxidative stability of krill oils were evaluated.

### 3.3. Comparison of the Accelerated Oxidative Stability of Krill Oils Added with Antioxidants at Different Time Points

The POV values of all krill oils went up significantly, accompanied by the increase in storage time at 60 °C (Figure 2A), showing that all the oils were progressively oxidized (*p* < 0.05). Post to 2, 4, 6, and 8 days of storage, the POV values of TP-D (added with TP during the extraction process), TP-TPP-D (added with TP-TPP during the extraction process), TP-A (added with TP after the extraction process) and TP-TPP-A (added with TP-TPP after the extraction process) were significantly lower than that of the control group without adding any antioxidants, indicating that these antioxidants could significantly retard the primary oxidation of krill oils (*p* < 0.05). Importantly, adding antioxidants during the extraction process is more effective than adding antioxidants after the extraction process. For example, after 4 days of storage, the POV values of the control, TP-D, TP-TPP-D, TP-A, and TP-TPP-A groups were 0.46, 0.42, 0.40, 0.44, and 0.42 mmol/kg, respectively. Apparently, the order of inhibitory ability was: TP-TPP-D > TP-TPP-A > TP-D, TP-A > Con (*p* < 0.05).

The TBARS values of all krill oils went up significantly, accompanied by the increase in storage time at 60 °C (Figure 2B), showing that the generation of secondary oxidation products during accelerated storage (*p* < 0.05). Post to 2, 4, 6, and 8 days of storage, the TBARS values of TP-D, TP-TPP-D, TP-A, and TP-TPP-A were significantly lower than that of the control group, indicating that these antioxidants could significantly retard the secondary oxidation of krill oils (*p* < 0.05). Importantly, adding antioxidants during the extraction process is more effective than adding antioxidants after the extraction process. For example, after 6 days of storage, the TBARS values of the control, TP-D, TP-TPP-D, TP-A, and TP-TPP-A groups were 1.58, 1.34, 1.28, 1.44, and 1.43 mg MDA/kg, respectively. Apparently, the order of inhibitory ability was: TP-TPP-D >TP-D >TP-TPP-A, TP-A > Con (*p* < 0.05).

In order to further confirm the above-mentioned results, the changing trends in the fatty acid composition of krill oils added with antioxidants were measured (Table 3). After 8 days of storage, the PUFA values of the control, TP-D, TP-TPP-D, TP-A, and TP-TPP-A groups were 18.89, 23.49, 25.66, 22.73, and 24.11%, respectively. Apparently, the PUFA values of all krill oils went up significantly, accompanied by the increase in storage time at 60 °C (*p* < 0.05). By contrast, the PUFA values of krill oils added with antioxidants were significantly higher than that of the control group, while SFA and MUFA were lower than those of the control group (*p* < 0.05). Importantly, adding antioxidants during the extraction process is more effective than adding antioxidants after the extraction process. Meanwhile, TP-TPP-D exerted the best antioxidant effect.

The lipid composition was selected to measure the changing trends of free fatty acid (FFA) of krill oils added with antioxidants (Table 4). After 8 days of storage, the FFA values of the control, TP-D, TP-TPP-D, TP-A, and TP-TPP-A groups were 0.85, 0.73, 0.68, 0.77, and 0.76%, respectively. Apparently, the FFA values of all krill oils went up significantly, accompanied by the increase in storage time at 60 °C (*p* < 0.05). By contrast, the FFA values of krill oils added with antioxidants were significantly lower than that of the control group (*p* < 0.05). Importantly, adding antioxidants during the extraction process is more effective than adding antioxidants after the extraction process. Similar to the results of POV, TBARS, and the fatty acids composition, TP-TPP-D exerted the best antioxidant effect.

The above results clearly showed that adding the antioxidants at different time points of processing and storage could influence the oxidative stability of krill oils. By contrast, adding antioxidants during the extraction process is more effective than adding antioxidants after the extraction process. Obviously, TP-TPP-D exerted the best antioxidant effect.

## 4. Discussion

The formation of primary and secondary oxidation products occurs throughout the complicated process of lipid oxidation [27]. The POV was determined in order to monitor the primary oxidation products in this research. With the method [22], Fe (II) ions are oxidized to Fe (III) ions by oil hydroperoxides. Next, xylenol orange combines with Fe (III) ions to form a complex compound that has a peak absorbance of 560 nm. In this investigation, TBARS was taken into consideration to detect secondary oxidation products. Ketones, hydroxy compounds, aldehydes, epoxides, polymers, and oligomers are the byproducts of lipid secondary oxidation. Among them, the most commonly used labeling compound is MDA. After the reaction of MDA, colored trimethadione was formed, and the maximum absorption peak was at 532 nm [28].

PUFAs such as EPA and DHA can combine with oxygen to trigger the oxidation of lipids, which is carried out by a series of free radical reactions [11]. PUFAs lose an atom of hydrogen and produce lipid free radicals when they are exposed to initiators such as metal ions, light/ionizing radiation, heat, and metalloproteins. The lipid then reacts with the ground oxygen molecules to form peroxyl radicals, which form hydroperoxides and new lipid radicals. This process, a free radical chain reaction, can be repeated several times to generate lipid autoxidation [11,29]. Thus, ketones and aldehydes are produced as a result of the decomposition of PUFAs.

In this study, POV, TBARS, fatty acid composition, and lipid composition were used to evaluate the oxidative stability of krill oils added with single antioxidants or composite antioxidants. Our results indicated that the POV, TBARS, and FFA values of krill oils added with antioxidants were significantly lower than those of the control group without adding any antioxidants, while the PUFA contents of krill oils added with antioxidants were significantly higher than those of the control group without adding any antioxidants. Apparently, these antioxidants could significantly retard the oxidation of krill oils during extraction and storage. By contrast, TP and TP-TPP exerted the best antioxidant effect. Similar results have also been reported by other researchers. For example, Bai et al. reported that among the several novel natural single antioxidants (dihydromyricetin (DMY), phytic acid (PA), paeonol (PAE), propolis (PR), AOB, RE, TP, VE), TP significantly prevented tree peony seed oil from oxidation at the concentration of 0.04% (*w*/*w*) [30]. Moreover, Pei et al. reported that the sample of walnut oil with 100 mg/kg TP and 450 mg/kg TPP demonstrated the highest level of stability [31]. These results showed that among all the composite antioxidant mixtures, the TP-TPP demonstrated the strongest antioxidant ability. This is likely due to the fact that lipid oxidation in oils also produces several minor components, such as polar compounds and free fatty acids [32]. There is a ton of evidence that these components can interact with the tiny amounts of water in oils to form physical structures, which may be the location of lipid oxidation [33,34]. TP is more hydrophilic than TPP, and as a result, it has a stronger affinity for the interface of association colloids [33].

Our results indicated that the POV, TBARS, and FFA values of krill oils added with antioxidants during the extraction process were significantly lower than those of the krill oils added with antioxidants after the extraction process, while the PUFA contents of the former krill oil samples added with antioxidants during the extraction process were significantly higher than those of the latter krill oil samples added with antioxidants after the extraction process. Apparently, antioxidants added during the extraction process are more effective than those added after the extraction process. As is known, the oils, including soybean oil, colleseed oil, and Antarctic krill oil, are extracted by organic solvents. Obviously, oils are easily oxidized and degraded in the procedures of settling (contact with air) and evaporation (relatively high temperature). In addition, the extraction solvent of vegetable oil is No. 6 solvent (n-hexane), while the extraction solvent of krill oil is ethanol. Water in the krill meal can be easily extracted by using ethanol as an extraction solvent. During the extraction process, the EPA and DHA are more easily oxidized and degraded when water exists in the ethanol extract of krill oil. Therefore, adding antioxidants to the extraction solvent during the oil extraction process may possibly inhibit oil oxidation.

It is widely known that some vegetable oils, such as sesame oil and hemp seed oil, contain abundant amounts of natural antioxidant components. Many pieces of research have shown that these natural antioxidant components could effectively protect oils from oxidation. For example, Shen et al. suggested that in the seed oil extracted from seeds of three *Chenopodium* (red, white, and black) with hexane, black quinoa seed oil contained the highest content of PUFA [35]. On the one hand, there are inherent differences between the oils and fats of the raw seeds of different species. On the other hand, the natural antioxidant components synergistically extracted during the oil extraction process also directly affect the oil quality. Indeed, the tocopherol and phytosterols content of black quinoa seed oil was significantly high those of white quinoa seed oil and red quinoa seed oil. Moreover, Nehdi et al. reported that compared to the stripped seed oils, the nonstripped seed oils exhibited greater stability at about 60 °C [36]. This is mainly due to stripped seed oils being devoid of any tocopherols. Stripped seed oils remove minor components that act as antioxidants to prevent the oxidation of unsaturated fatty acids [37]. Surely, single antioxidants or composite antioxidants added during the extraction are more effective in inhibiting oil oxidation than those added after the extraction.

## 5. Conclusions

The results of the accelerated storage experiment at 60 °C showed that the composite antioxidants (TP-TPP) consisting of tea polyphenol (TP) and tea polyphenol palmitate (TPP) had an excellent antioxidant effect on Antarctic krill *(Euphausia superba)* oil. Importantly, adding TP-TPP into ethanol solvent during the extraction process is more effective than adding it to krill oil after the extraction process.

## Figures and Tables

**Figure 1 foods-11-03768-f001:**
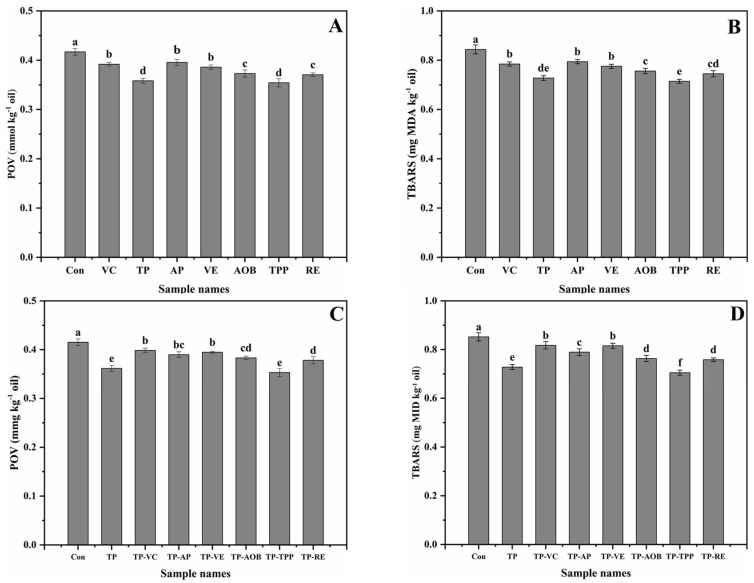
Changes of POV and TBARS of krill oils added with different single antioxidants (**A**,**B**) and composite antioxidants (**C**,**D**) during the extraction process. Con was the control krill oil without adding any antioxidants; VC, TP, AP, VE, AOB, TPP and RE were the krill oils added with vitamin C (VC), tea polyphenol (TP), ascorbyl palmitate (AP), vitamin E (VE), antioxidant of bamboo leaves (AOB), tea polyphenol palmitate (TPP) and rosemary extract (RE), respectively; TP-VC, TP-AP, TP-VE, TP-AOB, TP-TPP and TP-RE were the krill oils added with the binary mixtures comprised of TP and one of the other six antioxidants (VC, AP, VE, AOB, TPP and RE), respectively. All experiments were repeated three times. Different letters (a–f) indicate significant differences from each other (*p* < 0.05).

**Figure 2 foods-11-03768-f002:**
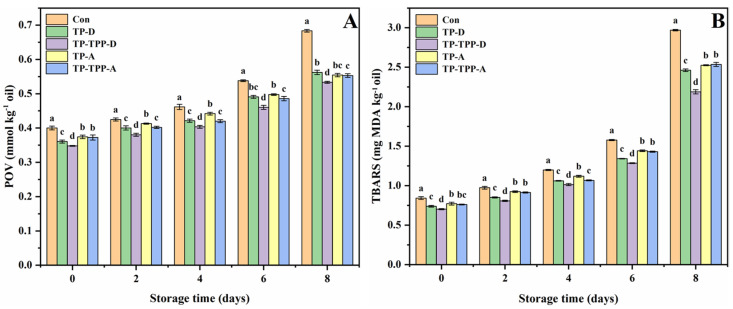
Changes of POV (**A**) and TBARS (**B**) of krill oils added with different antioxidants in the accelerated storage at 60 °C. Con was the control krill oil without adding any antioxidants; TP-D and TP-TPP-D were the krill oils added with the tea polyphenol (TP) and the binary mixtures of TP with tea polyphenol palmitate (TPP) during the extraction process, respectively; TP-A and TP-TPP-A were the krill oils added with the TP and the binary mixtures of TP with TPP after the extraction process, respectively. All experiments were repeated three times. Different letters (a–d) indicate significant differences from each other at same storage time (*p* < 0.05).

**Table 1 foods-11-03768-t001:** The FA composition (relative content, %) of krill oils added with different single antioxidants during the extraction process.

FA	Con	VC	TP	AP	VE	AOB	TPP	RE
C14:0	13.61 ± 0.03a	13.68 ± 0.12a	13.42 ± 0.29a	13.62 ± 0.28a	13.41 ± 0.11a	13.65 ± 0.28a	13.68 ± 0.18a	13.38 ± 0.14a
C16:0	28.75 ± 0.06a	27.96 ± 0.31a	26.31 ± 0.10a	28.37 ± 0.31a	27.34 ± 0.09a	27.25 ± 0.23a	27.14 ± 0.23a	27.13 ± 0.16a
C16:1	5.10 ± 0.10a	5.11 ± 0.12c	5.12 ± 0.09e	5.01 ± 0.04b	5.22 ± 0.20d	5.11 ± 0.09d	5.08 ± 0.09d	5.20 ± 0.09d
C17:0	3.43 ± 0.10ab	3.52 ± 0.08a	3.19 ± 0.10c	3.34 ± 0.09abc	3.42 ± 0.02a	3.30 ± 0.04bc	3.22 ± 0.12c	3.31 ± 0.11bc
C18:0	2.43 ± 0.13a	2.29 ± 0.04a	2.35 ± 0.11a	2.58 ± 0.24a	2.39 ± 0.15a	2.43 ± 0.11a	2.47 ± 0.18a	2.36 ± 0.02a
C18:2 n-6c	5.27 ± 0.17a	5.31 ± 0.17a	5.72 ± 0.19b	5.50 ± 0.01ab	5.51 ± 0.09ab	5.63 ± 0.24b	5.56 ± 0.16ab	5.64 ± 0.18b
C18:1 n-9c	20.85 ± 0.12a	20.47 ± 0.18abc	20.15 ± 0.12c	20.43 ± 0.35bc	20.62 ± 0.13ab	20.51 ± 0.21abc	18.22 ± 0.20d	20.26 ± 0.19bc
C20:0	1.52 ± 0.07ab	1.51 ± 0.04ab	1.60 ± 0.01ab	1.47 ± 0.12a	1.49 ± 0.04ab	1.56 ± 0.16ab	1.78 ± 0.06c	1.66 ± 0.06bc
C20:1	1.27 ± 0.08a	1.31 ± 0.10a	1.30 ± 0.05a	1.21 ± 0.12a	1.39 ± 0.09a	1.36 ± 0.04a	1.31 ± 0.01a	1.25 ± 0.05a
C18:3 n-3	1.32 ± 0.04a	1.42 ± 0.06ab	1.53 ± 0.09c	1.40 ± 0.02ab	1.49 ± 0.06bc	1.53 ± 0.06c	1.46 ± 0.04bc	1.40 ± 0.05ab
C20:2	1.25 ± 0.04a	1.46 ± 0.09b	1.56 ± 0.11b	1.41 ± 0.11ab	1.41 ± 0.06ab	1.40 ± 0.06ab	1.48 ± 0.15b	1.40 ± 0.14ab
C20:5 n-3	10.60 ± 0.25a	11.05 ± 0.16abc	11.78 ± 0.46c	10.84 ± 0.02ab	10.99 ± 0.21abc	11.29 ± 0.16bc	13.24 ± 0.38d	11.37 ± 0.26c
C22:6 n-3	4.60 ± 0.08a	4.92 ± 0.15ab	5.97 ± 0.23d	4.79 ± 0.10a	5.33 ± 0.18bc	5.38 ± 0.39c	5.36 ± 0.15c	5.64 ± 0.31cd
SFA	49.73 ± 0.20a	48.95 ± 0.41abc	47.04 ± 0.17b	49.38 ± 0.19b	48.05 ± 0.21ab	48.19 ± 0.70abc	48.29 ± 0.29d	47.84 ± 0.46bc
MUFA	27.22 ± 0.15a	26.90 ± 0.06abc	26.60 ± 0.11c	26.66 ± 0.27c	27.23 ± 0.16ab	26.98 ± 0.26abc	24.61 ± 0.17d	26.71 ± 0.16bc
PUFA	23.04 ± 0.30a	24.15 ± 0.38bc	26.36 ± 0.28g	23.96 ± 0.11b	24.73 ± 0.35cd	25.23 ± 0.32de	27.11 ± 0.41h	25.45 ± 0.38f

Con was the control krill oil without adding any antioxidants; VC, TP, AP, VE, AOB, TPP and RE were the krill oils added with vitamin C (VC), tea polyphenol (TP), ascorbyl palmitate (AP), vitamin E (VE), antioxidant of bamboo leaves (AOB), tea polyphenol palmitate (TPP) and rosemary extract (RE) during the extraction process, respectively. All experiments were repeated three times. Different letters (a–h) in the same row indicate significant differences from each other (*p* < 0.05). Abbreviations: FA, fatty acid; SFA, saturated fatty acid; MUFA, monounsaturated fatty acid; PUFA, polyunsaturated fatty acid.

**Table 2 foods-11-03768-t002:** The FA composition (relative content, %) of krill oils added with different composite antioxidants during the extraction process.

FA	Con	TP	TP-VC	TP-AP	TP-VE	TP-AOB	TP-TPP	TP-RE
C14:0	13.65 ± 0.08a	13.25 ± 0.14b	13.65 ± 0.11a	13.39 ± 0.06ab	13.54 ± 0.13ab	13.51 ± 0.25ab	13.46 ± 0.22ab	13.45 ± 0.15ab
C16:0	28.64 ± 0.17a	26.87 ± 0.17de	27.78 ± 0.11b	27.25 ± 0.25cd	27.53 ± 0.21bc	27.23 ± 0.24cd	26.74 ± 0.23e	27.23 ± 0.23cd
C16:1	5.22 ± 0.21ab	4.98 ± 0.05c	5.27 ± 0.06a	5.26 ± 0.08a	5.27 ± 0.14a	5.17 ± 0.11abc	5.04 ± 0.03bc	5.11 ± 0.07abc
C17:0	3.36 ± 0.13ab	3.19 ± 0.08cd	3.38 ± 0.07ab	3.39 ± 0.07ab	3.47 ± 0.13a	3.30 ± 0.07abc	3.06 ± 0.07d	3.22 ± 0.05bc
C18:0	2.37 ± 0.20a	2.34 ± 0.07ab	2.38 ± 0.07a	2.37 ± 0.03a	2.34 ± 0.10ab	2.35 ± 0.04a	2.15 ± 0.12b	2.38 ± 0.10a
C18:2 n-6c	5.30 ± 0.20a	5.91 ± 0.17e	5.41 ± 0.08ab	5.64 ± 0.09bcd	5.49 ± 0.03abc	5.73 ± 0.15cde	5.56 ± 0.12abc	5.82 ± 0.16de
C18:1 n-9c	20.80 ± 0.21a	19.95 ± 0.11c	20.67 ± 0.24a	20.78 ± 0.10a	20.67 ± 0.20a	20.32 ± 0.22b	18.39 ± 0.20d	20.05 ± 0.08bc
C20:0	1.54 ± 0.05a	1.54 ± 0.04a	1.51 ± 0.10a	1.50 ± 0.05a	1.47 ± 0.06a	1.50 ± 0.09a	1.81 ± 0.14b	1.50 ± 0.10a
C20:1	1.32 ± 0.03b	1.37 ± 0.09ab	1.41 ± 0.06ab	1.33 ± 0.04b	1.46 ± 0.09a	1.33 ± 0.03b	1.16 ± 0.06c	1.32 ± 0.06b
C18:3 n-3	1.37 ± 0.05b	1.44 ± 0.01ab	1.40 ± 0.04ab	1.51 ± 0.06a	1.40 ± 0.04ab	1.45 ± 0.01ab	1.08 ± 0.03c	1.47 ± 0.14ab
C20:2	1.28 ± 0.05a	1.55 ± 0.10d	1.43 ± 0.09bcd	1.36 ± 0.03abc	1.39 ± 0.01bc	1.50 ± 0.11cd	1.33 ± 0.05a	1.41 ± 0.11bcd
C20:5 n-3	10.55 ± 0.30a	11.74 ± 0.31c	10.94 ± 0.17ab	11.08 ± 0.18b	11.06 ± 0.10b	11.28 ± 0.24bc	14.26 ± 0.49d	11.29 ± 0.24bc
C22:6 n-3	4.60 ± 0.14a	5.87 ± 0.24d	4.77 ± 0.09ab	5.13 ± 0.16bc	4.91 ± 0.16ab	5.33 ± 0.33c	6.06 ± 0.16d	5.74 ± 0.28d
SFA	49.56 ± 0.31a	47.18 ± 0.38d	48.70 ± 0.14b	47.90 ± 0.19c	48.35 ± 0.22bc	47.89 ± 0.37c	47.22 ± 0.57d	47.79 ± 0.50cd
MUFA	27.34 ± 0.15a	26.30 ± 0.25c	27.35 ± 0.22a	27.37 ± 0.08a	27.40 ± 0.18a	26.82 ± 0.14b	24.59 ± 0.27d	26.48 ± 0.16c
PUFA	23.10 ± 0.32a	26.51 ± 0.51e	23.95 ± 0.31b	24.73 ± 0.26bc	24.26 ± 0.21b	25.29 ± 0.48cd	28.19 ± 0.70f	25.73 ± 0.50d

Con was the control krill oil without adding any antioxidants; TP was the krill oil added with tea polyphenol (TP) during the extraction process; TP-VC, TP-AP, TP-VE, TP-AOB, TP-TPP and TP-RE were the krill oils added with the binary mixtures comprised of TP and one of the other six antioxidants (vitamin C (VC), ascorbyl palmitate (AP), vitamin E (VE), antioxidant of bamboo leaves (AOB), tea polyphenol palmitate (TPP) and rosemary extract (RE)) during the extraction process, respectively. All experiments were repeated three times. Different letters (a–f) in the same row indicate significant differences from each other (*p* < 0.05). Abbreviations: FA, fatty acid; SFA, saturated fatty acid; MUFA, monounsaturated fatty acid; PUFA, polyunsaturated fatty acid.

**Table 3 foods-11-03768-t003:** The FA composition (relative content, %) of krill oils added with different antioxidants in the accelerated storage at 60 °C.

FA	0-Day					8-Day				
	Con	TP-D	TP-TPP-D	TP-A	TP-TPP-A	Con	TP-D	TP-TPP-D	TP-A	TP-TPP-A
C14:0	13.78 ± 0.28A	14.29 ± 1.09A	13.46 ± 0.55A	14.21 ± 0.59A	13.20 ± 0.49A	14.53 ± 0.05a	14.46 ± 0.47a	14.56 ± 0.30a	14.33 ± 0.28a	14.73 ± 0.38a
C16:0	28.89 ± 0.26A	25.64 ± 0.91C	26.61 ± 0.32BC	27.06 ± 0.93B	27.28 ± 0.26B	29.40 ± 0.33a	28.11 ± 0.41b	26.05 ± 0.10d	29.06 ± 0.55a	27.13 ± 0.22c
C16:1	5.18 ± 0.09A	5.85 ± 0.24B	5.09 ± 0.52A	5.40 ± 0.29AB	5.42 ± 0.12AB	5.67 ± 0.08a	5.40 ± 0.46a	5.63 ± 0.10a	5.41 ± 0.32a	5.18 ± 0.22a
C17:0	3.26 ± 0.11AB	3.05 ± 0.41A	3.01 ± 0.18A	3.61 ± 0.23B	3.11 ± 0.04A	3.75 ± 0.18a	3.48 ± 0.06b	3.87 ± 0.04a	3.66 ± 0.11ab	3.48 ± 0.15b
C18:0	2.46 ± 0.15A	2.21 ± 0.12BC	2.17 ± 0.06C	2.37 ± 0.08AB	2.28 ± 0.07ABC	2.89 ± 0.06a	2.40 ± 0.16b	2.47 ± 0.05b	2.36 ± 0.25b	2.63 ± 0.12ab
C18:2 n-6c	5.43 ± 0.01A	5.72 ± 0.20AB	5.66 ± 0.10AB	5.85 ± 0.25B	5.75 ± 0.30AB	4.97 ± 0.08a	5.56 ± 0.12bc	6.08 ± 0.29d	5.36 ± 0.31ab	5.96 ± 0.25cd
C18:1 n-9c	20.47 ± 0.44A	19.04 ± 0.46BC	18.39 ± 0.46C	19.14 ± 0.63BC	19.56 ± 0.34B	20.73 ± 0.22a	19.58 ± 1.29b	18.45 ± 0.09b	19.32 ± 0.39b	19.48 ± 0.32b
C20:0	1.51 ± 0.11A	1.92 ± 0.31B	1.73 ± 0.07AB	1.56 ± 0.12A	1.77 ± 0.02AB	1.97 ± 0.05a	1.84 ± 0.37a	1.96 ± 0.10a	1.79 ± 0.09a	1.83 ± 0.11a
C20:1	1.32 ± 0.01A	1.25 ± 0.13A	1.11 ± 0.08B	1.33 ± 0.05A	1.34 ± 0.05A	2.17 ± 0.10a	1.25 ± 0.09b	1.35 ± 0.02b	1.34 ± 0.20b	1.44 ± 0.01b
C18:3 n-3	1.25 ± 0.12A	1.07 ± 0.11A	1.09 ± 0.16A	1.05 ± 0.05A	1.16 ± 0.16A	0.91 ± 0.08a	1.19 ± 0.12b	1.00 ± 0.05a	0.99 ± 0.06a	1.01 ± 0.06a
C20:2	1.25 ± 0.04AB	1.32 ± 0.19AB	1.17 ± 0.10A	1.16 ± 0.17A	1.47 ± 0.18B	1.00 ± 0.06a	1.20 ± 0.03b	1.05 ± 0.09a	1.38 ± 0.09c	1.29 ± 0.09bc
C20:5 n-3	10.63 ± 0.26A	12.36 ± 0.19B	14.08 ± 0.13C	12.40 ± 0.46B	12.14 ± 0.50B	8.23 ± 0.25a	11.02 ± 0.04b	12.63 ± 0.91c	10.04 ± 0.78b	10.98 ± 0.52b
C22:6 n-3	4.57 ± 0.01A	6.28 ± 0.66B	6.46 ± 0.77B	4.87 ± 0.28A	5.53 ± 0.44AB	3.78 ± 0.24a	4.52 ± 0.47b	4.91 ± 0.25b	4.96 ± 0.58b	4.88 ± 0.29b
SFA	49.90 ± 0.70A	47.11 ± 0.84B	46.96 ± 0.12B	48.94 ± 0.06A	47.64 ± 0.67B	52.54 ± 0.36a	50.29 ± 0.83bc	48.91 ± 0.41d	51.20 ± 0.59b	49.79 ± 0.71cd
MUFA	26.97 ± 0.35A	26.14 ± 0.38AB	24.59 ± 0.99C	25.86 ± 0.36B	26.31 ± 0.29AB	28.57 ± 0.17a	26.23 ± 0.96b	25.43 ± 0.20b	26.07 ± 0.25b	26.10 ± 0.19b
PUFA	23.13 ± 0.35A	26.75 ± 0.71C	28.45 ± 1.01D	25.20 ± 0.42B	26.05 ± 0.79BC	18.89 ± 0.38a	23.49 ± 0.44bc	25.66 ± 0.53d	22.73 ± 0.76b	24.11 ± 0.89c

Con was the control krill oil without adding any antioxidants; TP-D and TP-TPP-D were the krill oils added with the tea polyphenol (TP) and the binary mixtures of TP with tea polyphenol palmitate (TPP) during the extraction process, respectively; TP-A and TP-TPP-A were the krill oils added with the TP and the binary mixtures of TP with TPP after the extraction process, respectively. All experiments were repeated three times. Different upper case letters (A–D) and lower case letters (a–d) in the same row indicate significant differences from each other at same storage time (*p* < 0.05). Abbreviations: FA, fatty acid; SFA, saturated fatty acid; MUFA, monounsaturated fatty acid; PUFA, polyunsaturated fatty acid.

**Table 4 foods-11-03768-t004:** The lipid composition (relative content, %) of krill oils added with different antioxidants in the accelerated storage at 60 °C.

Time	Sample Names	TG	FFA	DG	Cho	MG	PL
0-day	Con	44.21 ± 0.24A	0.66 ± 0.01D	2.24 ± 0.18A	1.68 ± 0.07A	0.31 ± 0.02A	50.90 ± 0.36B
	TP-D	44.59 ± 0.18AB	0.60 ± 0.01B	2.48 ± 0.20A	1.78 ± 0.06A	0.32 ± 0.04A	50.23 ± 0.08A
	TP-TPP-D	44.47 ± 0.24AB	0.53 ± 0.01A	2.48 ± 0.06A	1.84 ± 0.14A	0.35 ± 0.02A	50.32 ± 0.38A
	TP-A	44.66 ± 0.13B	0.62 ± 0.01C	2.22 ± 0.27A	1.83 ± 0.12A	0.35 ± 0.01A	50.32 ± 0.31A
	TP-TPP-A	44.44 ± 0.18AB	0.61 ± 0.02BC	2.23 ± 0.26A	1.85 ± 0.14A	0.32 ± 0.02A	50.54 ± 0.18AB
8-day	Con	44.47 ± 0.42a	0.85 ± 0.01d	2.73 ± 0.24b	1.82 ± 0.16a	0.34 ± 0.01a	49.80 ± 0.40a
	TP-D	44.72 ± 0.15a	0.73 ± 0.01b	2.58 ± 0.21ab	1.95 ± 0.06a	0.34 ± 0.01a	49.68 ± 0.03a
	TP-TPP-D	44.53 ± 0.08a	0.68 ± 0.02a	2.71 ± 0.08b	1.80 ± 0.23a	0.34 ± 0.01a	49.94 ± 0.23a
	TP-A	44.54 ± 0.36a	0.77 ± 0.02c	2.35 ± 0.05a	1.80 ± 0.13a	0.36 ± 0.03a	50.18 ± 0.29a
	TP-TPP-A	44.52 ± 0.10a	0.76 ± 0.01bc	2.39 ± 0.11a	1.94 ± 0.09a	0.35 ± 0.01a	50.04 ± 0.26a

Con was the control krill oil without adding any antioxidants; TP-D and TP-TPP-D were the krill oils added with the tea polyphenol (TP) and the binary mixtures of TP with tea polyphenol palmitate (TPP) during the extraction process, respectively; TP-A and TP-TPP-A were the krill oils added with the TP and the binary mixtures of TP with TPP after the extraction process, respectively. All experiments were repeated three times. Different upper case letters (A–D) and lower case letters (a–d) in the same column indicate significant differences from each other with the same storage time (*p* < 0.05). Abbreviations: TG, triglyceride; FFA, free fatty acid; DG, diglyceride; Cho, cholesterol; MG, monoglyceride; PL, phospholipid.

## Data Availability

Data are contained within the article.

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
