# Peer review of "Effects of Tea Polyphenol and Its Combination with Other Antioxidants Added during the Extraction Process on Oxidative Stability of Antarctic Krill (Euphausia superba) Oil"

_foods, 2022, doi:10.3390/foods11233768_

Round 1
Reviewer 1 Report
1- In page 2 lines 73-75 : Each of these abbreviations AP, VC, VE, AOB, RE and TPP in " Given this, this study aimed to select the most effective single antioxidant (vitamin C, VC; tea polyphenol, TP; ascorbyl palmitate, AP; vitamin E, VE; antioxidant of bamboo leaves, AOB; tea polyphenol palmitate, TPP; rosemary extract, RE) "must be added separate brackets to facilitate the reader .
2- Page 3 line 98: The solvent was removed at 30°C in a rotary evaporator how?
3- Page 3 line 101-102 : Especially, based on the oil extraction rate, the single antioxidant (VC, TP, AP, VE, 101 AOB, TPP or RE) was added to the ethanol at its maximum allowable quantity allowed (the maximum allowable quantity should be added for each antioxidant)
4- Page 3 lines118-119 : The krill oil samples added with antioxidants during or after the extraction process, 118 were taken at regular intervals of 2 days until 8 days during an accelerated storage experiment at 60°C. why 60°? Are you made optimum conditions!
5- Page 4 line 146: the gas chromatographic (GC) analysis program should be written
6- Page 12 lines 395-405: Conclusions is not clear to understand , it should be rewritten!
Author Response
Dear reviewer:
Re: foods-2013907
We are grateful to you for constructive comments and some important changes requested in our manuscript. We have accordingly carefully revised our manuscript to address all the comments made. The main revisions are explained below as “Response to Reviewer 1’s Comments”. We trust that our manuscript is now ready for further processing for publication in the “Foods”.
Reviewer #1:
Comments:
- In page 2 lines 73-75: Each of these abbreviations AP, VC, VE, AOB, RE and TPP in " Given this, this study aimed to select the most effective single antioxidant (vitamin C, VC; tea polyphenol, TP; ascorbyl palmitate, AP; vitamin E, VE; antioxidant of bamboo leaves, AOB; tea polyphenol palmitate, TPP; rosemary extract, RE) "must be added separate brackets to facilitate the reader .
Response: We were really sorry for our careless mistakes. According to your guideline, "Given this, this study aimed to select the most effective single antioxidant (vitamin C, VC; tea polyphenol, TP; ascorbyl palmitate, AP; vitamin E, VE; antioxidant of bamboo leaves, AOB; tea polyphenol palmitate, TPP; rosemary extract, RE) and the most effective composite antioxidant, as well as compare the accelerated oxidative stability of krill oils added with antioxidants at different time points (during or after the extraction process). " has been changed to "Given this, this study aimed to select the most effective single antioxidant and composite antioxidant among vitamin C (VC), tea polyphenol (TP), ascorbyl palmitate (AP), vitamin E (VE), antioxidant of bamboo leaves (AOB), tea polyphenol palmitate (TPP), rosemary extract (RE) and their binary mixtures, as well as compare the accelerated oxidative stability of krill oils added with antioxidants at different time points (during or after the extraction process)." marked with font colors. Please read Page 2, Lines 73-78.
- Page 3 line 98: The solvent was removed at 30°C in a rotary evaporator how?
Response: Thank you very much for your carefully review. In order to facilitate the understanding, "Subsequently, the solvent was removed at 30°C in a rotary evaporator." has been changed to "Subsequently, the filtrate was collected and the ethanol in filtrate was removed through rotary evaporation at 30°C." marked with font colors. Please read Page 3, Lines 99-100.
Especially, the rotary evaporator is shown as follows:
- Page 3 line 101-102: Especially, based on the oil extraction rate, the single antioxidant (VC, TP, AP, VE, 101 AOB, TPP or RE) was added to the ethanol at its maximum allowable quantity allowed (the maximum allowable quantity should be added for each antioxidant).
Response: We are appreciative of the reviewer’s suggestion. According to your guideline, "Especially, based on the oil extraction rate, the single antioxidant (VC, TP, AP, VE, AOB, TPP or RE) was added to the ethanol at its maximum allowable quantity allowed by Chinese Standard GB 2760-2014 [21]." has been changed to "Especially, based on the oil extraction rate, the single antioxidant (VC, TP, AP, VE, AOB, TPP or RE) was added to the ethanol at its maximum allowable quantity (maq) allowed by Chinese Standard GB 2760-2014 [21]. The maq values of VC, TP, AP, VE, AOB, TPP and RE were 0.2, 0.6, 0.2, 0.4, 0.5, 0.6 and 0.7 g/kg oil, respectively." marked with font colors. Please read Page 3, Lines 102-105.
- Page 3 lines118-119: The krill oil samples added with antioxidants during or after the extraction process, 118 were taken at regular intervals of 2 days until 8 days during an accelerated storage experiment at 60°C. why 60°C? Are you made optimum conditions!
Response: Thank you very much for your carefully review. Lipids present in oils are susceptible to oxidation, which affects food quality and safety. Given this, oxidative stability is an important parameter when evaluating the quality of oils and fats, as it gives a good estimation of their susceptibility to oxidative deterioration. In general, the oxidative stability of different kinds of oils was studied using the Schaal Oven test (60°C) method based on determination of peroxide value and Rancimat method based on conductometric measurements. Especially, the accelerated storage condition of 60°C is optimal for estimation of the shelf life at 25°C wherein 1day at 60°C is equal to 8.79 days at 25°C. Therefore, 60°C has been widely used in many accelerated storage experiments. Thank you again for your carefully review.
- Page 4 line 146: the gas chromatographic (GC) analysis program should be written.
Response: Thank you very much for your carefully review. According to your guideline, the related analysis program of "FAME separation was performed by using a Supelco SP 2560 capillary column (100 m × 0.25 mm, 0.2 μm). The injection volume was 1 μL with a split ratio of 20:1, and the injector temperature was set as 220°C. The FID temperature was set as 260°C, and the constant carrier gas (N2) flow was set as 2.0 mL/min. The heating procedure is as follows: 120°C for 9 min; increasing (20°C/min) to 200°C and held for 5 min; increasing (3°C/min) to 230°C and held for 10 min. All fatty acids were identified by comparing their retention times with those of the standards [25]." has been added in the revised manuscript marked with font colors. Please read Page 4, Lines 149-155.
- Page 12 lines 395-405: Conclusions is not clear to understand, it should be rewritten!
Response: Thank you very much for your carefully review. According to your guideline, "In the present study, based on the analysis of peroxide value (POV), thiobarbituric acid reactive substances (TBARS), fatty acid composition and lipid class composition, the results indicated that tea polyphenol (TP) exerted significantly higher antioxidant effectiveness than others including vitamin C (VC), ascorbyl palmitate (AP), vitamin E (VE), antioxidant of bamboo leaves (AOB), tea polyphenol palmitate (TPP) and rosemary extract (RE). Moreover, TP-TPP exerted the most excellent antioxidant effect among the TP and the composite antioxidants (TP-TPP, TP-RE, TP-AOB, TP-AP, TP-VE and TP-VC) combining the TP with the other commonly used antioxidants (VC, AP, VE, AOB, TPP and RE). Importantly, adding TP-TPP into ethanol solvent during the extraction process, is more effective than adding it into krill oil after the extraction process." has been changed to "The results of accelerated storage experiment at 60°C showed that the composite antioxidants (TP-TPP) consist of tea poly-phenol (TP) and tea polyphenol palmitate (TPP) had excellent antioxidant effect on Antarctic krill (Euphausia superba) oil. Importantly, adding TP-TPP into ethanol solvent during the extraction process, is more effective than adding it into krill oil after the extraction process." in the revised manuscript marked with font colors. Please read Page 12, Lines 406-410.

Reviewer 2 Report
The manuscript deals with experimental tests to evaluate the oxidative stability of krill oils added with the single antioxidants or the composite antioxidants. This paper is well written and the data clearly organized.
Line 131. I suggest to the authors add the values of maximum allowable quantity
Line 158. I suggest the authors to describe which compounds will be analyzed
Line 311 – I suggest the authors compare the krill oil class composition data with other data in the literature, mainly concerning the high phospholipid content of the oil
Line 334: “absorption peak was at 532 nm” and in line 136 for Thiobarbituric methodology the authors mentioned 560 nm. Please, check which one is correct.
Line 395. I suggest the authors to add more information to the conclusion with more details obtained from the study
Author Response
Dear reviewer:
Re: foods-2013907
We are grateful to you for constructive comments and some important changes requested in our manuscript. We have accordingly carefully revised our manuscript to address all the comments made. The main revisions are explained below as “Response to Reviewer 2’s Comments”. We trust that our manuscript is now ready for further processing for publication in the “Foods”.
Reviewer #2:
Comments:
- Line 131. I suggest to the authors add the values of maximum allowable quantity
Response: We are appreciative of the reviewer’s suggestion. According to your guideline, "Especially, based on the oil extraction rate, the single antioxidant (VC, TP, AP, VE, AOB, TPP or RE) was added to the ethanol at its maximum allowable quantity allowed by Chinese Standard GB 2760-2014 [21]." has been changed to "Especially, based on the oil extraction rate, the single antioxidant (VC, TP, AP, VE, AOB, TPP or RE) was added to the ethanol at its maximum allowable quantity (maq) allowed by Chinese Standard GB 2760-2014 [21]. The maq values of VC, TP, AP, VE, AOB, TPP and RE were 0.2, 0.6, 0.2, 0.4, 0.5, 0.6 and 0.7 g/kg oil, respectively." marked with font colors. Please read Page 3, Lines 102-105.
- Line 158. I suggest the authors to describe which compounds will be analyzed
Response: Thank you very much for your carefully review. According to your guideline, "By dividing the peak area of the separated lipid by the sum of the peak areas of all the separated lipids, the lipid class composition was computed." has been changed to "By dividing the peak area of the separated lipid by the sum of the peak areas of all the separated lipids, the lipid class compositions of triglyceride (TG), free fatty acid (FFA), diglyceride (DG), cholesterol (Cho), monoglyceride (MG) and phospholipid (PL) were obtained." marked with font colors. Please read Page 4, Lines 165-168.
- Line 311 – I suggest the authors compare the krill oil class composition data with other data in the literature, mainly concerning the high phospholipid content of the oil.
Response: We are appreciative of the reviewer’s suggestion. Free fatty acid (FFA) content is a marker of oils hydrolysis. Compared with neutral oil, FFA is less stable, and is more prone to oxidation and to turning rancid. Therefore, FFA is a key indicator of quality and commercial value of oils. Especially, the oxidation hydrolysis of phospholipids will also produce FFAs. However, the magnitude of phospholipid content is usually not very significant. Given this, we wanted to clarify the effects of antioxidants on the production of FFA, and the lipid class composition was measured to reflect the change of FFA. Thank you again for your carefully review.
- Line 334: “absorption peak was at 532 nm” and in line 136 for Thiobarbituric methodology the authors mentioned 560 nm. Please, check which one is correct.
Response: We were really sorry for our careless mistakes. Accordingly, "After cooling and centrifuging at 3000 g for 10 minutes, took 200 uL of the mixture's upper layer, and measured the absorbance at 560 nm." has been changed to "After cooling and centrifuging at 3000 g for 10 minutes, took 200 uL of the mixture's upper layer, and measured the absorbance at 532 nm." marked with font colors. Please read Page 3, Lines 137-139.
- Line 395. I suggest the authors to add more information to the conclusion with more details obtained from the study.
Response: We are appreciative of the reviewer’s suggestion. According to your guideline, the section of conclusion has been changed to "The results of accelerated storage experiment at 60°C showed that the composite antioxidants (TP-TPP) consist of tea poly-phenol (TP) and tea polyphenol palmitate (TPP) had excellent antioxidant effect on Antarctic krill (Euphausia superba) oil. Importantly, adding TP-TPP into ethanol solvent during the extraction process, is more effective than adding it into krill oil after the extraction process." in the revised manuscript marked with font colors. Please read Page 11, Lines 406-410.

Round 2
Reviewer 1 Report
non